# The Inhibition Effects of Sodium Nitroprusside on the Survival of Differentiated Neural Stem Cells through the p38 Pathway

**DOI:** 10.3390/brainsci13030438

**Published:** 2023-03-03

**Authors:** Lingling Jiao, Tongying Xu, Xixun Du, Xi Chen, Qian Jiao, Hong Jiang

**Affiliations:** 1Department of Physiology, Shandong Provincial Key Laboratory of Pathogenesis and Prevention of Neurological Disorders, State Key Disciplines: Physiology, Shandong Provincial Collaborative Innovation Center for Neurodegenerative Disorders, School of Basic Medicine, Qingdao University, Qingdao 266071, China; 2College of Health and Life Science, University of Health and Rehabilitation Sciences, Qingdao 266071, China

**Keywords:** neural stem cells, sodium nitroprusside, survival, differentiation

## Abstract

Nitric oxide (NO) is a crucial factor in regulating neuronal development. However, certain effects of NO are complex under different physiological conditions. In this study, we used differentiated neural stem cells (NSCs), which contained neural progenitor cells, neurons, astrocytes, and oligodendrocytes, to observe the physiological effects of sodium nitroprusside (SNP) on the early developmental stage of the nervous system. After SNP treatment for 24 h, the results showed that SNP at 100 μM, 200 μM, 300 μM, and 400 μM concentrations resulted in reduced cell viability and increased cleaved caspase 3 levels, while no significant changes were found at 50 μM. There were no effects on neuronal differentiation in the SNP-treated groups. The phosphorylation of p38 was also significantly upregulated with SNP concentrations of 100 μM, 200 μM, 300 μM, and 400 μM, with no changes for 50 μM concentration in comparison with the control. We also observed that the levels of phosphorylation increased with the increasing concentration of SNP. To further explore the possible role of p38 in SNP-regulated survival of differentiated NSCs, SB202190, the antagonist of p38 mitogen-activated protein kinase, at a concentration of 10 mM, was pretreated for 30 min, and the ratio of phosphorylated p38 was found to be decreased after treatment with SNP. Survival and cell viability increased in the SB202190 and SNP co-treated group. Taken together, our results suggested that p38 is involved in the cell survival of NSCs, regulated by NO.

## 1. Introduction

Nitric oxide (NO), created from L-arginine and oxygen by nitric oxide synthases (NOSs), is vital for neuronal development. Under certain specific physiological conditions, NO can both promote and inhibit the proliferation and differentiation of neural stem cells (NSCs) [1,2].

NO is implicated in the regulation of neuro-axonal growth and synaptogenesis in the developing brain and has an important role in long-term structural and functional processes of neuronal circuits in the adult brain [3]. Physiological amounts of NO are required to preserve neuronal survival and health, and conversely, a deficiency of NO may exacerbate neuropathology [4]. The studies of Scholz [5] and Hindley [6] suggested that the regulation of nerve development by NO plays a role, through information transmission of soluble guanylate cyclase (sGC), in the cyclic guanosine monophosphate (cGMP) pathway. In a study of Drosophila melanogaster, NO mediated the transformation of stem cells from proliferation to differentiation by inhibiting DNA synthesis [7]. Endothelial nitric oxide synthase (eNOS) is a key regulatory factor in the development of the fetal craniofacial skeleton. Fetal dural cells and calcaneal osteoblasts express eNOS, and eNOS-derived NO enhances the proliferation and differentiation of both types of cells [8]. In NSCs, brain-derived neurotrophic factor (BDNF) may facilitate NO production by boosting neuronal nitric oxide synthase (nNOS) expression in differentiated neurons, resulting in a shift from proliferation to differentiation of neural progenitor cells (NPCs), and NO produced by nNOS expression in neurons could affect the fate of neighboring NPCs, causing NPCs to exit the cell cycle [3]. Apart from the influence of nNOS in NSCs, the expression of inducible nitric oxide synthase (iNOS) in NSCs results in the synthesis of a large amount of NO, which also leads to the transformation of NSCs from proliferation to differentiation [9]. Champlin confirmed that exogenous NO could inhibit the proliferation of NSCs in brain slices cultured in vitro [10]. However, other evidence has suggested that this conclusion is controversial [11]. Exogenous NO inhibits neurogenesis and its inhibitory effect on nNOS and iNOS, respectively [1]. On the other hand, iNOS inhibitors can reduce pathological NO produced by reactive microglia and NSCs [12]. Therefore, inhibition of pathological NO production by iNOS may exert neuroprotective and protective effects against brain injury [13,14]. This has multiple implications for promoting nerve regeneration and improving central nervous system disorders or injuries.

In the early stage of brain development, the central nervous system contains various cell types, such as NSCs, NPCs, neurons, astrocytes, and oligodendrocytes. As a whole network, the nervous system receives internal environment signals and accordingly makes changes. The effect and mechanism of NO on the neural network during development need further exploration. SNP is a donor of NO, and its ability to produce NO depends on the structure contained in an iron core surrounded by five cyanide ion molecules and one nitrosonium ion (NO^+^) [15,16]. In the present study, a system containing multiple cell types differentiated from NSCs was used to investigate the effects of different concentrations of NO on the survival and differentiation of NSCs. Our results showed that the survival rate of differentiated NSCs decreased with an increasing concentration of SNP, and the survival rate of differentiated NSCs increased after administration of the p38 antagonist SB202190, indicating that NO affects the survival of differentiated NSCs through p38.

## 2. Methods and Materials

### 2.1. Isolation and Culture of Cerebral Cortex NSCs

NSCs were isolated from the embryonic day 14–15 (E14–15) cerebral cortex of a rat. The meninges were carefully removed. Then, the cerebral cortex tissue was trypsinized and mechanically triturated into single cells and cultured in a serum-free growth medium as in our previous study [17,18]. The NSCs growth medium contained DMEM/F12 (VivaCell Biosciences, Cat#C3130-0500, Shanghai, China), 1% N2 supplement (Gibco, Cat#A1370701, Grand Island, NY, USA), 2% B27 supplement (Gibco, Cat#17504044, Grand Island, NY, USA), 10 ng/mL bFGF (Peprotech, Cat#100-18B, Wuhan, China), 20 ng/mL EGF (Gibco, Cat#PHG0315, Grand Island, NY, USA), 100 U/mL penicillin, 100 μg/mL streptomycin (Solarbio, Cat#P1400, Beijing, China), and 2.5 μg/mL heparin (Sigma, Cat#H3149, St. Louis, MO, USA). Primary cell cultures were sub-cultured every 5–7 days. The differentiation medium contained DMEM/F12, 1% fetal bovine serum (FBS, Gibco, Cat#10099-141, Grand Island, NY, USA), 1% N2 supplement, 2% B27 supplement, 100 U/mL penicillin, and 100 μg/mL streptomycin. All procedures that involved laboratory work with animals were in accordance with the ethical guidelines of the NIH Regulations for Experimentation on Laboratory Animals and Qingdao University regulations (ethical approval code: QDU-AEC-2023027).

### 2.2. Cell Viability Assay

SNP (Sigma, Cat#71778, St. Louis, MO, USA) was dissolved in a 0.9% sodium chloride solution. Equal numbers of NSCs (about 10,000) were suspended in 100 μL differentiation medium and added into each well of 96-well plates. NSCs were treated with SNP at concentrations of 50, 100, 200, 300, and 400 μmol/L (μM) for 24 h. A cell counting kit-8 (CCK-8, Targetmol, Cat#C0005, Boston, MA, USA) was used to analyze cell viability. Ten μL reagent from CCK-8 were added into each well and incubated for 2 h, following the protocol. WST-8 in CCK-8 reagent was reduced by dehydrogenases in mitochondria to produce an orange-colored product (formazan). The number of living cells affects the amount of formazan dye. Cell viability was observed at 450 nm with a SpectraMax Plus 384 Microplate Reader (Molecular Devices, San Jose, CA, USA).

SB202190 (Sigma, Cat#S0767, St. Louis, MO, USA) was dissolved in DMSO. The differentiated NSCs were prepared as above. Cells were pretreated with the antagonist SB202190 at a concentration of 10 µM for 30 min and then treated with SNP at 200 μM for 24 h.

### 2.3. Immunofluorescence Staining

Cultured neurospheres were stained by nestin to confirm the features of NSCs. NSCs were plated on Poly-L-Lysine (Sigma, Cat#P1399, St. Louis, MO, USA)-coated coverslips and fixed with 4% paraformaldehyde (PFA) (Hushi, Cat#80096618, Shanghai, China) at room temperature and then blocked with a solution containing 5% normal goat serum (Solarbio, Cat#SL038, Beijing, China) and 0.25% Triton X-100 (Biosharp, Cat#BS084, Beijing, China) in phosphate-buffered saline (PBS, Thermo Fisher Scientific, Waltham, MA, USA). Primary antibodies diluted in 0.01 M PBS were incubated overnight at 4 °C in a humidified chamber, including monoclonal mouse anti-nestin (1:200, Millipore, Cat#MAB353, Boston, MA, USA), monoclonal mouse anti-β-tubulin III (1:200, Millipore, Cat#MAB5564, Boston, MA, USA), and monoclonal mouse anti-glial fibrillary acidic protein (GFAP) (1:1000, Millipore, Cat#MAB360, Boston, MA, USA), which were used to identify NSCs, neurons, and astrocytes, respectively. After coverslips were rinsed three times with 0.01 M PBS, the secondary antibodies were incubated at room temperature for 2 h. TRITC- and FITC-conjugated goat anti-mouse IgG (1:500; Invitrogen, Cat#A31570, Cat#A21202, Grand Island, NY, USA) were used as secondary antibodies. Cell nuclei were counterstained with DAPI containing mounting media (Beyotime, Cat#C1006, Shanghai, China) and visualized under a fluorescent microscope (Carl Zeiss AX10, Aalen, Germany). For the negative control, the primary antibody was replaced by a blocking buffer.

### 2.4. Western Blotting Analysis

The protein levels of β-tubulin III and GFAP were quantified by immunoblotting to investigate the neuronal and glial differentiation of NSCs. In addition, the protein levels of p38 and p-p38, caspase 3, and cleaved caspase 3 were also detected to identify the related mechanisms of cell behavior changes. Primary antibodies monoclonal mouse anti-β-tubulin III (1:1000), rabbit polyclonal rabbit anti-p38 (1:1000, Cell Signaling Technology, Cat#9212S, Boston, MA, USA), polyclonal rabbit anti-p-p38 (1:1000, Cell Signaling Technology, Cat#9211S, Boston, MA, USA), rabbit polyclonal rabbit anti-caspase 3 (1:1000, Cell Signaling Technology, Cat#9662S, Boston, MA, USA), and monoclonal mouse anti-β-actin (1:5000, Bioss, Cat#bs-0061R, Beijing, China) were used. After SNP or SB202190 treatment for 24 h, NSCs were lysed in RIPA lysis buffer (Beyotime, Shanghai, China) with protease inhibitors. Cell lysates were centrifugalized at 12,000× *g* rpm for 15 min at 4 °C to remove insoluble material. Samples were subjected to electrophoresis using 10% SDS polyacrylamide gels (30% Acrylamide/Bis Soln: MD bio, Cat#F5290901, Shanghai, China; 10% APS: Boster Biological Technology, Cat#1166-10, Wuhai, China; SDS-PAGE Separating Gel Buffer (4X): Cowin Biotech, Cat#CW0026, Beijing, China; SDS-PAGE Stacking Gel Buffer (4X): Cowin Biotech, Cat#CW0025, Beijing, China; TEMED: MD bio, Cat#F1100225, Shanghai, China), and then transferred to nitrocellulose membranes. The nitrocellulose membranes were incubated in 5% non-fat dry milk in Tris-buffered saline with Tween (TBST) (Tris: BioFroxx, Cat#1115KG001, Einhausen, Germany; Glycine: BioFroxx, Cat#1275GR500, Einhausen, Germany; Tween-20: Solarbio Cat#T8220, Beijing, China) for 1 h, and then were incubated with primary antibodies at 4 °C overnight. After incubation with secondary antibodies at room temperature for 2 h, bands were detected by chemiluminescence with the ECL method (Pierce Biotechnology, Waltham, MA, USA). The luminescent signal was recorded using the Amersham ImageQuant 800 (Cytiva, Logan, UT, USA), and the data were analyzed with Image J (NIH).

### 2.5. Statistical Analysis

All data were analyzed with SPSS 17.0 software and Graphic Prism 9.0. The Student’s T-test was used to compare the differences between means in two groups. One-way ANOVA followed by Bonferroni multiple comparison tests was used to compare differences between means in more than two groups. *p* < 0.05 was considered significant.

## 3. Results

### 3.1. Culture and Identification of NSCs

Cells harvested from the rat embryonic cerebral cortex were cultured in DMEM/F12 serum-free growth medium and formed neurospheres after culturing for 4–5 days. Immunofluorescence staining showed that the cells in neurospheres were nestin-positive (Figure 1A). After 7 days of culturing in a differentiation medium, β-tubulin III positive neurons and GFAP-positive astrocytes were found in these cells (Figure 1B). The results suggested that the cells we cultured were NSCs that had proliferation and differentiation abilities.

### 3.2. SNP Decreased the Survival of Differentiated NSCs

To investigate the effect of SNP on the survival of differentiated NSCs, we treated the differentiated NSCs with SNP at concentrations of 50, 100, 200, 300, and 400 μM for 24 h. The phase contrast showed that the cells appeared brighter in the SNP 100, 200, 300, and 400 μM groups (Figure 2A,B). Cell viability was decreased in the SNP 200, 300, and 400 μM-treated groups. We detected the expression of caspase 3 and found that in comparison with the control group, the ratio of cleaved caspase 3 increased in the SNP 100, 200, 300, and 400 μM groups (Figure 2C,D).

### 3.3. SNP Affected the Survival of Differentiated NSCs through p38

In comparison with the control, the phosphorylation of p38 significantly increased in the SNP 100, 200, 300, and 400 μM groups (Figure 3A, *p* < 0.001). No significant difference was found in the phosphorylation level of p38 (Figure 3B, *p* > 0.05) in the SNP 50 μM group. To further explore the possible role of p38 in the SNP-regulated survival of NSCs, the NSC’s neuro-biological behaviors were re-analyzed after pretreatment with the antagonist SB202190 at a concentration of 10 µM for 30 min. The ratio of p-p38/p38 (Figure 3C,D), cell survival (Figure 3E), and cell viability (Figure 3F, *p* < 0.05) of differentiated NSCs were increased in the SB202190+SNP (200 μM) combination treatment group in comparison with the SNP (200 μM)-treated alone group.

### 3.4. SNP Had No Effect on the Neuronal Differentiation of NSCs

Neural progenitors still existed after NSCs differentiation for 3 days. Therefore, we treated the NSCs with 1 day of differentiation with SNP at different concentrations for 24 h to investigate the effect of SNP on the differentiation of NSCs. Immunofluorescence staining showed that the percentage of β-tubulin III positive cells did not change in the SNP-treated groups at different concentrations in comparison with the control (Figure 4A,B). The Western blotting results showed that the expression of β-tubulin III underwent no significant changes in the SNP groups in comparison with the control (Figure 4C,D).

## 4. Discussion

NSCs need to locate in a suitable microenvironment to keep self-renewing and self-maintaining [19,20]. If the signals from the microenvironment, such as the cytoarchitecture, extracellular matrix proteins, and soluble factors change, the neurobiological behaviors of NSCs will follow and adopt new physiological activities. Therefore, a better understanding of how NSCs are affected by changes in the microenvironment will provide insight into the neurobiology of NSCs in normal and diseased states.

Neurogenesis is accompanied by the proliferation of NSCs and neuronal differentiation in the brain. NO is an important factor in axon growth and guidance, synaptic plasticity, and the transition from proliferation to differentiation of NSCs [21]. NO can also diffuse freely across cell membranes, allowing it to act on neighboring neurons and glia. On the other hand, excessive production of NO has been identified as one of the major causes of the pathogenesis of many neurodegenerative diseases, such as Alzheimer’s disease and Parkinson’s disease [22,23]. High concentrations of NO induce endoplasmic reticulum stress through oxidative and nitrosative stress, leading to neuronal injury and activation of the mitochondrial apoptosis signaling pathway [24]. After cerebral ischemia, local excitatory amino acids increase, activating N-methyl-D-aspartate (NMDA) receptors and Ca^2+^ inflow. Excess NO can damage nerve cells and is involved in the NMDA-mediated neurotoxicity of primary cortical substances. NO has the function of inhibiting the growth, differentiation, and hemoglobinization of primary erythroid cells [25]. Studies have found that NO has different effects on different cells. The effects of NO on neurons can be complex and context-dependent. NO can enhance or inhibit neurotransmission, depending on the receptor subtypes and neuronal circuits involved [26]. NO can also enhance synaptic plasticity, a process that underlies learning and memory, by increasing the strength of synapses between neurons [27]. Activated glial cells in the nervous system can be activated by inflammatory chemokines. NO plays an important role in the occurrence and development of inflammation, as well as injury repair processes. In glia, NO has been shown to have a variety of effects depending on the type of glial cell. In astrocytes, NO can act as a signaling molecule to modulate calcium signaling and the release of gliotransmitters, which can, in turn, modulate synaptic transmission [28,29]. NO has been proven to promote oligodendrocyte differentiation and maturation by increasing the density of mature oligodendrocytes and myelin content in the immature rat brain [30]. This effect may be due to the ability of NO to modulate intracellular calcium levels, which are important for oligodendrocyte precursor cell differentiation [31]. NO has also been shown to play a role in the activation of microglia, the immune cells of the central nervous system, and can contribute to neuroinflammation in the context of injury or disease [32,33]. NO can also inhibit the proliferation and migration of vascular smooth muscle cells, promote the proliferation and migration of endothelial cells, and suppress endothelial cell apoptosis [34,35,36]. NO plays a role in promoting vascular regeneration and inhibiting vascular calcification by regulating the differentiation of endogenous vascular stem/progenitor cells into vascular endothelial cells and inhibiting their osteogenic differentiation [37]. These results suggest the specific effects of NO depend on the context and the type of cells involved, and dysregulation of NO signaling can contribute to a variety of neurological and psychiatric disorders.

The role of NO in neurogenesis is still controversial [38,39]. Some researchers think that NO is a physiological inhibitor of neurogenesis, while others believe that NO is beneficial to neurogenesis. For example, NO modulates hippocampal neurogenesis by independent extra- and intracellular signaling pathways [40]. Neuropeptide Y (NPY) mediates neuronal NO synthase and activates intracellular NO signaling, leading to an increase in cell proliferation through the NO/cGMP/cGMP-dependent protein kinase signaling pathway [40]. On the contrary, extracellular NO treatment can inhibit hippocampal neurogenesis by activating the extracellular signal-regulated kinase (ERK) 1/2 signaling pathway [40]. Moreover, evidence has shown that NO from an inflammatory origin resulted in nitration of the EGF receptors. The decreased function of the EGF receptors caused a downregulated activation of the ERK/MAPK pathway, which prevented the proliferation of NSCs [12]. In fact, the level of cellular NO needs to be precisely controlled because chronic NOS inhibition has a stimulatory effect on NSC proliferation [41]. NO has been shown to promote neuron migration in the developing brain. In the embryonic mouse brain, NO produced by the enzyme nNOS enhanced the migration of interneurons from the ganglionic eminences to the cortex [42]. NO has been shown to have both pro- and anti-differentiation effects on neurons, depending on the specific context and microenvironment. NO has been found to promote cell differentiation by inhibiting the synthesis of DNA and cell proliferation during neurodevelopment [9]. Studies have shown that NO/cGMP signaling enhances the differentiation of neural precursors derived from hESCs [43]. L-NAME, an inhibitor of NOS, inhibited the neuronal differentiation of NSCs induced by BDNF [3]. Low-dose NO has been reported to inhibit some differentiation markers that are expressed when bFGF is removed from the medium. In addition, NO has antiviral effects and dysregulates neuronal and glial differentiation, both in 2D cultures of NPCs and 3D cortical organoids [44]. NO decreased Sox2 levels and impaired mitochondrial function; both are important regulators of neural differentiation [44]. Zika virus-infected microglia promoted the expression of iNOS; the conditional medium of Zika virus-infected microglia showed inhibitory effects on cell proliferation and neuronal differentiation of NPCs [45]. In addition, high NO levels promoted axon pruning but repressed the regrowth of axons by inhibiting the physical interaction between the E75 and UNF nuclear receptors [46]. In contrast, overexpression of NOS from three independent transgenes did not significantly inhibit axon regrowth [46]. This evidence indicated that NO modulates a switching mechanism that affects neuronal remodeling and maturation. In addition to regulating the proliferation and differentiation of NSCs, NO has also been found to affect the migration of NSCs in the brain. NO has also been shown to promote the migration of neurons in the developing olfactory bulb [47]. In the process of nerve regeneration and migration after resection of the olfactory bulb and pituitary, the migration of nerve cells was positively correlated with the expression of NOS [48]. The expression changed in nNOS during neonatal rat cerebral cortical development, affecting neural migration and the projection of axons [49]. In order to further elucidate the effect of NO on the survival and differentiation of NSCs, we constructed a system in which NSCs were induced to differentiate in vitro for 7 days, containing multiple cell types. Studies have found that elevated NO concentrations led to a decrease in the survival rate of NSCs, and NO acts as a powerful oxidant that can destroy many types of biomolecules, especially in protein oxidation, lipid peroxidation, and mitochondrial dysfunction processes [50,51].

Mitogen-activated protein kinase (MAPK) signaling can be activated by a variety of cellular stimuli, and the three major subfamilies of MAPKs include c-Jun N-terminal kinase (JNK), ERK, and p38. As the effect of NO on NSCs is highly related to the activation of the MAPK signaling pathway, in this experiment, we treated the experimental group with the p38 antagonist SB202190 and found that, compared with the SNP group, the survival rate of the SB202190 combined treatment group with SNP was significantly increased. Therefore, we believe that p38 is involved in NO damage to NSCs. Previous research observed the death-enhancing effect of BDNF on NO-induced neural damage through p38 and ERK on primary cultured cortical neurons [52]. The cell type selected in our study was different from previous studies. Differentiated NSCs contain neural progenitor cells, neurons, astrocytes, and oligodendrocytes, which are more like the early developmental stage of the nervous system. Therefore, we observed the effect of SNP on the nervous system, rather than a single cell type, to identify the role of p38 in this process. We found that a concentration of SNP higher than 100 μM induced cell damage, which might be caused by a high level of NO. Previous studies reported that endogenous NO induces phosphorylation of p38 through NMDAR and mediates p38 signaling through NMDAR, stimulating stress (including oxidative stress) to promote cell death. Exogenous NO inhibits this process and exerts a neuroprotective effect. One bioactivity of NO is the control of the post-translational modifications through protein S-nitrosation [53]. S-nitrosation induced by NO results in endothelial hyperpermeability [54] and DNA damage [55]. In the present experiment, we only studied p38; the roles of JNK, ERK, and S-nitrosation in NO on NSCs are currently unknown. Therefore, studying these markers could allow us to better understand the microenvironment of NSCs and improve their survival rate.

## 5. Conclusions

The present study suggested that SNP mainly affects the survival of differentiated NSCs, without having a significant effect on neuronal differentiation. Furthermore, we identified that p38 might be involved in the cell survival of the differentiated NSCs regulated by NO.

## Figures and Tables

**Figure 1 brainsci-13-00438-f001:**
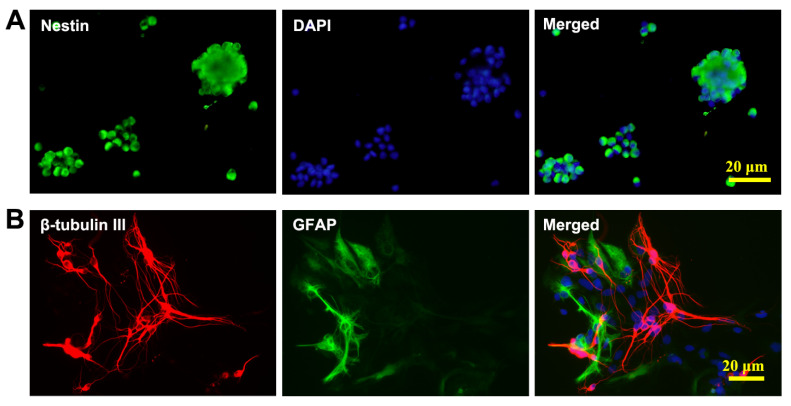
Culture and identification of NSCs. (**A**) Cells in spheres were nestin positive. (**B**) The β-tubulin III positive cells and GFAP positive cells were observed after 7 days of culture in the differentiation medium. Scale bars = 20 μm.

**Figure 2 brainsci-13-00438-f002:**
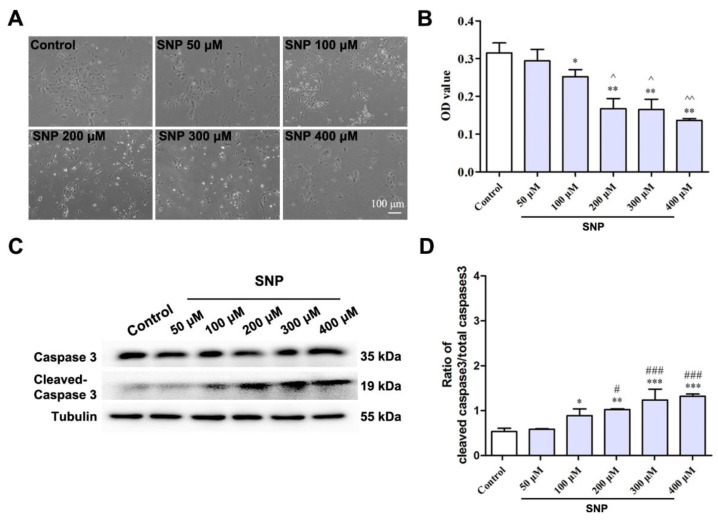
The effects of SNP on the survival of differentiated NSCs. (**A**) The survival and morphological changes of differentiated NSCs after SNP treatment. Scale bar = 100 μm. (**B**) Cell survival of differentiated NSCs after SNP treatment. (**C**) Expressions of caspase 3 and cleaved caspase 3 in differentiated NSCs. (**D**) Data analysis of the ratio of cleaved caspase 3/total caspase 3. The values of the ratio of cleaved caspase 3/total caspase 3 in SNP-treated groups were normalized by the control. All data were obtained from three independent experiments and more than three replicates for every experiment. The values are the mean ± SD and were analyzed via a one-way ANOVA test. Compared with control * *p* < 0.05, ** *p* < 0.01, *** *p* < 0.001; compared with the SNP 50 μM group ^#^
*p* < 0.05, ^###^
*p* < 0.001; compared with the SNP 100 μM group ^^^
*p* < 0.05, ^^^^
*p* < 0.01; compared with the SNP 200 μM group; compared with the SNP 300 μM group, *n* = 5.

**Figure 3 brainsci-13-00438-f003:**
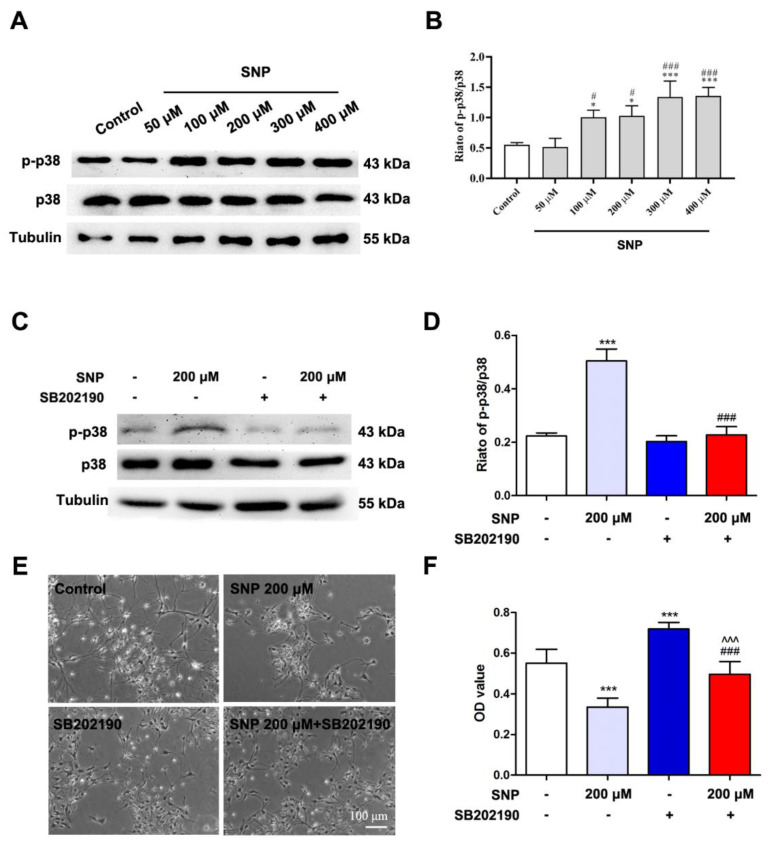
Effects of p38 on the survival of differentiated NSCs induced by SNP. (**A**,**B**) The expression of p38 and p-p38 of differentiated NSCs after SNP treatment. All data were obtained from three independent experiments and more than three replicates for every experiment. The values are the mean ± SD and were analyzed via a one-way ANOVA test. Compared with control * *p* < 0.05, *** *p* < 0.001; compared with the SNP 50 μM group ^#^
*p* < 0.05, ^###^
*p* < 0.001, *n* = 3. (**C**,**D**) Expressions of p38 and p-p38 were detected after SNP and SB202190 treatment. (**E**) The survival and morphological changes of differentiated NSCs after SNP and SB202190 treatment. Scale bar = 100 μm. (**F**) Cell survival of differentiated NSCs after SNP and SB202190 treatment. All data of D and F were obtained from three independent experiments and more than three replicates for every experiment. The values are the mean ± SD and were analyzed via *t*-test. Compared with control *** *p* < 0.001; compared with the SNP 200 μM group ^###^
*p* < 0.001; compared with the SB202190 group ^^^^^
*p* < 0.001, *n* = 5.

**Figure 4 brainsci-13-00438-f004:**
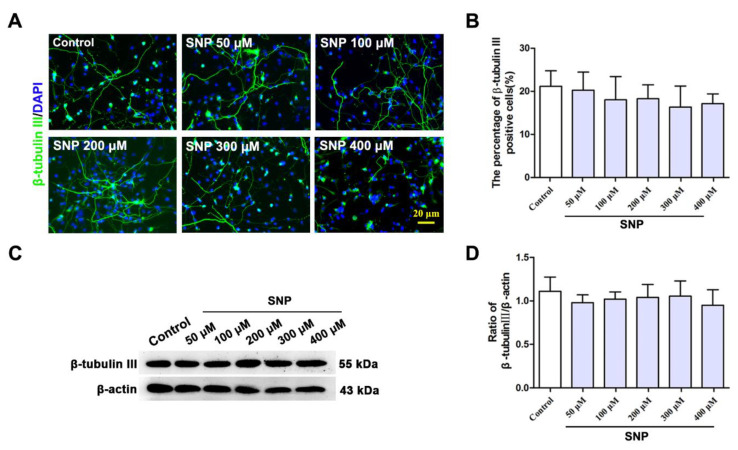
The effects of SNP on the differentiation of NSCs. (**A**) Immunocytochemistry staining of β-tubulin III positive cells. Scale bar = 50 μm. (**B**) Data analysis of the percentage of β-tubulin III positive cells, *n* = 5. (**C**) The expression of β-tubulin III was tested by Western blots. (**D**) Data analysis of the ratio of β-tubulin III/β-actin. All data were obtained from three independent experiments and three replicates for every experiment. The values are the mean ± SD and were analyzed via a one-way ANOVA test, *n* = 3.

## Data Availability

Not applicable.

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
