# Peer review of "The Inhibition Effects of Sodium Nitroprusside on the Survival of Differentiated Neural Stem Cells through the p38 Pathway"

_brainsci, 2023, doi:10.3390/brainsci13030438_

Round 1

Reviewer 1 Report

In this work the authors are investigating the effect of SNP on survival and differentiation of NSC derived from embryonic rat cortex and maintained in vitro. The SNP was tested at various concentrations as well as in combination with SB202190 (p38 MAPK inhibitor) since this pathway is known to be involved in nitric oxide-induced cell death and in general during neurodevelopment, axonal growth etc.    

The experimental design of this study is clear and simple, however, the results are relatively few and were not extensively described.

It is not clear what is the novelty and original contribution of this work respect to the previous observations, e.g. already in 2001, Cheng et al performed SNP treatment at similar concentrations and showed the increased cell death upon increasing concentration of SNP at three different time points (4, 7 and 10 hours). https://www.sciencedirect.com/science/article/pii/S0021925819829708

Importantly, in this work the duration of the treatment with SNP is not provided and sufficiently described for each set of experiments reported in the figures (methods, results and figure legends are missing this fundamental information; only for cell viability in the methods section it is reported that cells were analysed after 24h but this is not known for Western blot experiments). It is also not reported if the washes were performed or if the drug was left in the cell medium before the Immunocytochemistry or Western blot experiments. If the authors use different inhibitor, different cell line, different treatment (in terms of duration, concentration etc) or different assay to measure cell response, compared to the already published data, then this should be clearly specified.

In the same work by Cheng et al (2001) it was already demonstrated that p38 was involved and that its inhibition (by SB203580) abolished NO-induced cell death.

Overall the correct use of English language should be respected in describing scientific results. The extensive correction/rephrasing is required.

Abstract

Line 13: This sentence should be rephrased: NSC cannot differentiate into NSC (this is defined as self-renewal) but they can differentiate into neurons, astrocytes and oligodendrocytes.

Line 16: Define the time at which this is observed (minutes/hours/days after treatment)

Line 17: correct the part „of in SNP group“ – unclear meaning

Line 17-18: Rephrase the following sentence in order to be clear:

„The phosphorylation of p38 significantly increased in SNP 100 μM, 200 μM, 300 μM and 400 μM treated groups.“

As it is written it is not clear if the phosphorylation increases for all treated groups respect to the control or if it linearly increases with the increasing concentration of SNP. It is fundamental to describe properly the obtained results.

Line 19-20: the inhibitor should be described – it is not clear which protein is inhibited by the action of SB202190

„at 10µM concentration“ instead of  „with concentration 10µM”

“were pretreated” instead of “were pretreatment”

 p-p38 – define that this refers to phosphorilated form (respect to the p38)

„after treatment with“ (instead of  „after treated by“)

Line 22: remove „that“ from the sentence

Introduction

Line 27: „plays“ (instead of „play“)

Line 28: rephrase the sentence („...inhibition or promotion on neural stem cells (NSCs) neurobiological behaviors...“ – unclear meaning)

Line 33 and 35: NO (instead of „no“); „.“ is missing at the end of the sentences.

Line 37: explain the full names of SGC/cGMP pathway and explain its role

Line 40: the sentence „Fetal dural cells and calvarial osteoblasts express endothelial nitric oxide synthase, and endothelial nitric oxide synthase derived nitric oxide enhances the proliferation and differentiation of fetal dural cells and calvarial osteoblasts“ should be rewritten (excessive repeating). The abbreviations (NO, NOS) should be used as they were defined already at the beginning of the Introduction.

Line 44-49: define BDNF, nNOS, NPCs, iNOS at first mention

Line 51: Surname (Champlin) instead of the first name (David) should be used when citing other people's work

Line 60: correct the following sentence avoiding unnecessary repeating:“... the nerve system contained various cell types contained NSCs, progenitor cells...“ What is the difference between NSC and progenitor cells? Do you consider progenitor cells more committed population respect to the NSC? Overlapping terminology exists (stem and progenitor cells are often distinguished) and this should be clarified to avoid misunderstanings.

Line 65: correct/rephrase this part of the sentence: „...coexisted with multiple cell types was used...“

Line 67: unclear sentence („the increase of NO donor SNP concentration“???)

Methods

Line 74: insert „in“ between „as“ and „our“ (as in our previous study)

Line 79: Primary cells or primary cell cultures (instead of „Primary cultured cells“)

Line 88: treated (instead of „treatment“)

Line 93-94: The producers of Poly-L-Lysine, PFA, NGS and Triton X-100 should be provided.

Line 113: add the producer of RIPA lysis buffer

Line 115: provide the source of SDS polyacrylamide gels (home-made or ready-for-use precast gels)

Line 117: provide the producer for TBST

Results

Line 131: „were found“ (instead of „could find“)

Line 132: NSC (instead of neural stem cells, since the abbreviated from was already defined previously)

Line 135: correct the figure legend for the part „spheres were almost nestin positive“ – what does it mean „almost positive“? As it is written it seems that the nestin staining is uncertain

Line 136: use greek symbol (β) for „beta“ instead of „b“ (since this is how it was reported in the main text)

Line 140: „at“ (instead of „in“)

Line 141: It is not explained sufficiently how the „refractivity“ of the cells changes at different concentration of SNP. How it was measured? If this was done only by visual inspection, then at least the basic observation should be explained (What it means if cells appear brighter or darker when observed by phase-contrast?)

Line 142-143:

-      At the end of the sentence Figure 2B should be cited in the main text.

-      „in comparison“ (instead of „in compared“) –this has to be corrected in many other parts of the main text (line 183, 185 etc)

-      „ratio“ (instead of „ration“)

Line 147: „change“ or „changes“ (instead of „changed“); rephrase the sentence describing panel B (e.g. Cell-viability analysis) avoiding unneccessary repeating („survival of differentiated NSCs“ is already the title the Figure 2.

Line 161: check the concentration (mM?) – micro-molar (µM) concentration was used for all the previously described treatments -is this a typing error?

Line 162: rewrite the sentence describing clearly the obtained results („in compared with SNP group“????)

Line 164: „in comparison“ (instead of „in compared“)

Line 182: add „treated“ after SNP

Discussion

Line 194-198: correct the language („NSCs are need... to keep self-renew and self-maintain“: „If the signal...changed...“) and explain better and into more detail the meaning of „cell structures“ as „signal of microenvironment“  

Line 213: „has two sides“ – it should be properly rephrased

Line 224: separate the two sentences (...cell types,studies...)

Line 231: use the abbraviation for SNP in order to be consistent throughout the text

Author Response

In this work the authors are investigating the effect of SNP on survival and differentiation of NSC derived from embryonic rat cortex and maintained in vitro. The SNP was tested at various concentrations as well as in combination with SB202190 (p38 MAPK inhibitor) since this pathway is known to be involved in nitric oxide-induced cell death and in general during neurodevelopment, axonal growth etc.    

The experimental design of this study is clear and simple, however, the results are relatively few and were not extensively described.

It is not clear what is the novelty and original contribution of this work respect to the previous observations, e.g. already in 2001, Cheng et al performed SNP treatment at similar concentrations and showed the increased cell death upon increasing concentration of SNP at three different time points (4, 7 and 10 hours).https://www.sciencedirect.com/science/article/pii/S0021925819829708

Response: Many thanks to your kindly suggestion. We carefully compared the differences between the present study and the peer research mentioned here (PMID: 11555660). Peer research observed the effects of NO on enhancing apoptosis of neural progenitor cells. The underlying mechanism might be related to p38 and ERK. The biggest difference between two studies is cell type selected. We used the differentiated NSCs which contained neural progenitor cells, neurons, astrocytes, and oligodendrocytes, which likes the early development stage of nerve system. Because the cells interact with each other in nervous system, we observed the effect of SNP on nervous system rather than a single cell type. The difference is discussed in this revision and marked with blue color.

Importantly, in this work the duration of the treatment with SNP is not provided and sufficiently described for each set of experiments reported in the figures (methods, results and figure legends are missing this fundamental information; only for cell viability in the methods section it is reported that cells were analyzed after 24h but this is not known for Western blot experiments). It is also not reported if the washes were performed or if the drug was left in the cell medium before the Immunocytochemistry or Western blot experiments. If the authors use different inhibitor, different cell line, different treatment (in terms of duration, concentration etc) or different assay to measure cell response, compared to the already published data, then this should be clearly specified.

 Response: Following your kindly suggestion, we provided the description for each set of experiments both in methods and results sections and marked with blue color.

In the same work by Cheng et al (2001) it was already demonstrated that p38 was involved and that its inhibition (by SB203580) abolished NO-induced cell death.

Response: Thanks for your comments. The role of NO on neurogenesis is still controversial. NO may have the bidirectional regulation of neurogenesis. Even Cheng et al (PMID: 11555660) was already demonstrated that p38 was involved in NO-induced neural progenitor cells death, while the effects of NO on the early stage development of nerve system is unclear. Considering the cells interaction and signaling tansimition with each other in nervous system, we observed the effect of SNP on the differentiated NSCs rather than a single cell type.

Overall the correct use of English language should be respected in describing scientific results. The extensive correction/rephrasing is required.

 Response: We greatly appreciated for your suggestions. The whole manuscript has been double checked thoroughly. The typos and the grammatical errors have been carefully corrected. All the corrections have been marked with blue in this revision.

Abstract

Line 13: This sentence should be rephrased: NSC cannot differentiate into NSC (this is defined as self-renewal) but they can differentiate into neurons, astrocytes and oligodendrocytes.

 Response: We appreciated very much for the professional suggestion. NSCs keep the ability of self-renewal. We revised the sentence as “In this study, we used the differentiated neural stem cells (NSCs) which contained neural progenitor cells, neurons, astrocytes, and oligodendrocytes to observe the physiological effects of sodium nitroprusside (SNP) on the early development stage of the nerve system”.

Line 16: Define the time at which this is observed (minutes/hours/days after treatment)

Response: Following your kindly suggestion, the time of observation points is added in this revision.

Line 17: correct the part „of in SNP group“ – unclear meaning 

Response: Thanks, this revision has been rewritten in accordance with your suggestions.

Line 17-18: Rephrase the following sentence in order to be clear: “The phosphorylation of p38 significantly increased in SNP 100 μM, 200 μM, 300 μM and 400 μM treated groups.” As it is written it is not clear if the phosphorylation increases for all treated groups respect to the control or if it linearly increases with the increasing concentration of SNP. It is fundamental to describe properly the obtained results.

Response: We appreciated your suggestion. We have been revised the sentence as follow: “ And the phosphorylation of p38 was also significantly up-regulated in SNP of 100 μM, 200 μM, 300 μM and 400 μM treated groups, and no changes in 50 μM in comparison with the control. We also observed that the levels of phosphorylation elevated with the concentration of SNP increases.”

Line 19-20: the inhibitor should be described – it is not clear which protein is inhibited by the action of SB202190

Response: We appreciated the suggestion, and we rewrote it as “To further explore the possible role of p38 in SNP-regulated survival of NSCs, SB202190, the antagonists of p38 mitogen-activated protein kinase…”

„at 10µM concentration“ instead of  „with concentration 10µM”

Response: Thank you for pointing these out, and we have corrected in this revision.

“were pretreated” instead of “were pretreatment”

Response: Thanks for your suggestion and we have corrected this typo, and carefully checked the manuscript to avoid similar issues.

 p-p38 – define that this refers to phosphorilated form (respect to the p38)

Response: Thank you for pointing these out, and we have corrected in this revision.

„after treatment with“ (instead of  „after treated by“)

Response: Thanks for your suggestion and we have corrected this typo, and carefully checked the manuscript to avoid similar issues.

Line 22: remove „that“ from the sentence

Response: Thank you for pointing this out, we have corrected it in this revision.

Introduction

Line 27: „plays“ (instead of „play“)

Response: Thanks you so much and we have corrected this typo, and carefully checked the manuscript to avoid similar issues.

Line 28: rephrase the sentence („...inhibition or promotion on neural stem cells (NSCs) neurobiological behaviors...“ – unclear meaning)

Response: Following your kindly suggestion, the sentence is revised as: Under certain specific physiological conditions, NO pairs can both promote and inhibit the proliferation and differentiation of neural stem cells (NSCs).

Line 33 and 35: NO (instead of „no“); „.“ is missing at the end of the sentences.

Response: We appreciated very much for the suggestion and corrected it in this typo.

Line 37: explain the full names of SGC/cGMP pathway and explain its role

Response: Thanks you so much and we have corrected this typo, and carefully checked the manuscript to avoid similar issues.

Line 40: the sentence „Fetal dural cells and calvarial osteoblasts express endothelial nitric oxide synthase, and endothelial nitric oxide synthase derived nitric oxide enhances the proliferation and differentiation of fetal dural cells and calvarial osteoblasts“ should be rewritten (excessive repeating). The abbreviations (NO, NOS) should be used as they were defined already at the beginning of the Introduction. √

Line 44-49: define BDNF, nNOS, NPCs, iNOS at first mention

Response: Thanks you so much and we have corrected this typo, and carefully checked the manuscript to avoid similar issues.

Line 51: Surname (Champlin) instead of the first name (David) should be used when citing other people's work

Response: Thank you for pointing this out, we have corrected it in this revision.

Line 60: correct the following sentence avoiding unnecessary repeating:“... the nerve system contained various cell types contained NSCs, progenitor cells...“ What is the difference between NSC and progenitor cells? Do you consider progenitor cells more committed population respect to the NSC? Overlapping terminology exists (stem and progenitor cells are often distinguished) and this should be clarified to avoid misunderstanings.

Response: We appreciated very much for the professional questions.

Line 65: correct/rephrase this part of the sentence: „...coexisted with multiple cell types was used...“

Response: Thank you for pointing this out, we have corrected it as “a system containing multiple cell types which differentiated from NSCs was used to investigate the effects of different concentrations of NO on the survival and differentiation of NSCs”.

Line 67: unclear sentence („the increase of NO donor SNP concentration“???)

Response: Thank you for pointing this out, we have corrected it as “Our results showed that the survival rate of differentiated NSCs decreased with the increasing concentration of SNP”.

Methods

Line 74: insert „in“ between „as“ and „our“ (as in our previous study)

Response: We are very sorry for mistake and we have changed “as in our previous study” in this revision.

Line 79: Primary cells or primary cell cultures (instead of „Primary cultured cells“)

Response: Thanks for your suggestion and we have corrected it in this typo.

Line 88: treated (instead of „treatment“)

Response: We are very sorry for mistake. “treatment” is revised as “treated”.

Line 93-94: The producers of Poly-L-Lysine, PFA, NGS and Triton X-100 should be provided. Line 113: add the producer of RIPA lysis buffer. Line 115: provide the source of SDS polyacrylamide gels (home-made or ready-for-use precast gels). Line 117: provide the producer for TBST

Response: We are sorry for the missing information. The missing information in materials was added and marked with blue color, and we carefully checked the manuscript to avoid similar issues.

Results

Line 131: „were found“ (instead of „could find“)

Response: Following your kindly suggestion, the term “were found” is stead of “could find” in the referred sentence.

Line 132: NSC (instead of neural stem cells, since the abbreviated from was already defined previously)

Response: We are very sorry for mistake. The abbreviation form of “NSCs” is instead of “neural stem cells” in the referred sentence.

Line 135: correct the figure legend for the part „spheres were almost nestin positive“ – what does it mean „almost positive“? As it is written it seems that the nestin staining is uncertain

Response: Following your kindly suggestion, the figure legend of Figure 1 is corrected in this revision.

Line 136: use greek symbol (β) for „beta“ instead of „b“ (since this is how it was reported in the main text)

Response: We are very sorry for mistake. “b” is replaced by symbol β in this revision.

Line 140: „at“ (instead of „in“)

Response: Following your kindly suggestion, the word “in” is stead of “at” in the referred sentence.

Line 141: It is not explained sufficiently how the „refractivity“ of the cells changes at different concentration of SNP. How it was measured? If this was done only by visual inspection, then at least the basic observation should be explained (What it means if cells appear brighter or darker when observed by phase-contrast?)

Response: We are very sorry for the confused description. The cell morphological alterations were observed by visual inspection. Cells appear brighter after SNP treatment when observed by phase-contrast. We added the information in results and marked with blue color.

Line 142-143:

-      At the end of the sentence Figure 2B should be cited in the main text.

Response: Following your kindly suggestion, Figure 2B was cited in this revision and marked with blue color.

-      „in comparison“ (instead of „in compared“) –this has to be corrected in many other parts of the main text (line 183, 185 etc)

Response: Following your kindly suggestion, the term “in comparison” is stead of “in compared” in the referred sentence.

-      „ratio“ (instead of „ration“)

Response: We are very sorry for mistake. “Ration” is revised as “ratio”.

Line 147: „change“ or „changes“ (instead of „changed“); rephrase the sentence describing panel B (e.g. Cell-viability analysis) avoiding unneccessary repeating („survival of differentiated NSCs“ is already the title the Figure 2.

Response: Following your kindly suggestion, the mentioned sentences are revised and marked with blue color in figure legend of Figure 2.

Line 161: check the concentration (mM?) – micro-molar (µM) concentration was used for all the previously described treatments -is this a typing error?

Response: We are very sorry for mistake. The concentration of SB202190 used in this present study is 10 µM.

Line 162: rewrite the sentence describing clearly the obtained results („in compared with SNP group“????)

Response: We are very sorry for the confused description. Following your kindly suggestion, the results in the part of “3.3 SNP affected the survival of differentiated NSCs through p38” were rewritten and marked with blue color.

Line 164: „in comparison“ (instead of „in compared“)

Response: Following your kindly suggestion, we used “in comparison” in this revision and marked with blue color.

Line 182: add „treated“ after SNP

Response: Following your kindly suggestion, “SNP-treated” is used in manuscript and marked with blue color.

Discussion

Line 194-198: correct the language („NSCs are need... to keep self-renew and self-maintain“: „If the signal...changed...“) and explain better and into more detail the meaning of „cell structures“ as „signal of microenvironment“ 

Response: We are very sorry for the confused description. The term “cell structures” is replaced by “cyto-architecture” in this revision and marked with blue color. Cyto-architecture contains the cell types and spatial position structure.

Line 213: „has two sides“ – it should be properly rephrased.

Response: Many thanks for the suggestion. The mentioned sentence is revised as “The role of NO on neurogenesis is still controversial” and marked with blue color.

Line 224: separate the two sentences (...cell types,studies...)

Response: Following your kindly suggestion, the mentioned sentence are separated into two sentences

Line 231: use the abbraviation for SNP in order to be consistent throughout the text .

Response: Many thanks for the suggestion. We used the abbreviation for SNP to be consistent throughout the text.

Reviewer 2 Report

The manuscript by Jiao et al. aimed to study the role of sodium nitroprusside (SNP) in regulating the survival and differentiation of neural stem cells (NSCs). The mechanistic findings in underlying signaling pathways will improve our understanding of physiology and pathophysiology of related neuronal diseases.

Here lists a few of my concerns that could improve the overall quality of the manuscript if addressed appropriately.

Q1: Major problem: The question (gap of knowledge), novelty, and importance of this study is not quite clear. The introduction and discussion was not satisfactory to give the readers a clear understanding of the previous findings by others and the new findings in this study. There is not enough evidence to show the unique of the current study. See Line 21, Line 198-199.

Q2: Introduction:

Since the authors chose SNP to study the role of nitric oxide (NO) in NSCs, it should associate the relation between SNP and NO in the introduction section, and present some background information from the literature how SNP played a role in NSCs related research, instead of purely talking about the NO. It’s aware that SNP and NO is different, and the entire manuscript is studying the role of SNP but not NO.

Q2: Method:

All the reagents being used in the manuscript need to be listed. There are missing items in Section 2. Methods and materials, such as p38 antagonist SB202190.

Q3: Figure 2:

The drug concentration dependent experiment commonly applies the drug at a log scale concentration dependent incubation in cells, instead of a linear scale concentration, such as Figure 2. Also, the range of the concentration needs to cover the IC50 of the drug, namely, including low concentration and high concentration to compare to the vehicle control. I would suggest the authors re-design the SNP concentration range to test in primary NSCs. For Fig 2B and 2D, all comparison should be pointed to the control, not the other drug concentration, thus, a Dunnett's test is preferred as the post-hoc analysis.

Q4: Figure 3:

Same as Figure 2, log scale of SNP incubation in cells should be applied, instead of the linear scale in Fig 3 A, B. For western blotting image presentation, although the authors are analyzing the ratio of p-p38 to p38, the internal control housekeeping gene expression should be shown in Fig 3 A, C. Importantly, the data in Fig 3 D and F graphs do not support the results described in Result section 3.3 Line 155-164. It shows that the pre-treatment with the p38 antagonist SB202190 has the same treatment effect as the SNP in increasing ratio of p-p38/p38 and decreasing OD value. Although the exact statistical significance is different, that does not change the interpretation of the results. Because p<0.05 was considered as significant, as described in Line 124. I would highly suggest the authors to test another p-38 antagonist, if the results are similar, then it may indicate that the SNP may not specifically work through the p-38 signaling in these NSCs.

Q5: Figure 4:

Same as Figure 2, log scale of SNP incubation in cells should be applied, instead of the linear scale in Fig 4.

Q6: Author names: first name should be in front of the last name.

Q7: When use the short name for nitric oxide, it should use capital letters “NO” instead of “no”. Please correct through the manuscript, such as Line 33: “Physiological amounts of no are required……” and others.

Q8: The sentence is too long and difficult for readers to understand, please consider breaking into small sentences, such as Line 40-43

Q9: Please use the full name when first time introducing a new term, such as “nNOS” (Line 45), “iNOS” (Line 49)

Q10: References needed to be cited, such as Line 54, 55

Q11: Grammar issues and typos needed to be corrected throughout the entire manuscript, such as Line 60-61, Line 131, Line 144

Author Response

The manuscript by Jiao et al. aimed to study the role of sodium nitroprusside (SNP) in regulating the survival and differentiation of neural stem cells (NSCs). The mechanistic findings in underlying signaling pathways will improve our understanding of physiology and pathophysiology of related neuronal diseases.

 Here lists a few of my concerns that could improve the overall quality of the manuscript if addressed appropriately.

  Response: We greatly appreciated for your suggestions. The whole manuscript has been double checked thoroughly. All the corrections have been marked with blue in this revision.

Q1: Major problem: The question (gap of knowledge), novelty, and importance of this study is not quite clear. The introduction and discussion was not satisfactory to give the readers a clear understanding of the previous findings by others and the new findings in this study. There is not enough evidence to show the unique of the current study. See Line 21, Line 198-199.

 Response: We greatly appreciate the reviewer’s comments. We used the differentiated NSCs which contained neural progenitor cells, neurons, astrocytes, and oligodendrocytes, which likes the early development stage of nerve system. Because the cells interact with each other in nervous system, we observed the effect of SNP on nervous system rather than a single cell type. The difference is discussed in this revision and marked with blue color.

Q2: Introduction:

Since the authors chose SNP to study the role of nitric oxide (NO) in NSCs, it should associate the relation between SNP and NO in the introduction section, and present some background information from the literature how SNP played a role in NSCs related research, instead of purely talking about the NO. It’s aware that SNP and NO is different, and the entire manuscript is studying the role of SNP but not NO.

 Response: Following your kindly suggestion, the relation between SNP and NO is added in the introduction section line 66-68 and marked with blue color. SNP is a donor of the NO. It consists of an iron core surrounded by five cyanide ion molecules and one molecule of the nitrosonium ion (NO+) [1]. Its capability of producing NO seems to depend on its interaction with sulfhydryl-containing molecules [2].

References:

  1. Zoupa, E. and N. Pitsikas, The Nitric Oxide (NO) Donor Sodium Nitroprusside (SNP) and Its Potential for the Schizophrenia Therapy: Lights and Shadows. Molecules, 2021. 26(11).
  2. Grossi, L. and S. D'Angelo, Sodium nitroprusside: mechanism of NO release mediated by sulfhydryl-containing molecules. J Med Chem, 2005. 48(7): p. 2622-6.

Q2: Method:

All the reagents being used in the manuscript need to be listed. There are missing items in Section 2. Methods and materials, such as p38 antagonist SB202190.

 Response: We are sorry for the missing information. The missing information in materials was added and marked with blue color, and we carefully checked the manuscript to avoid similar issues.

Q3: Figure 2:

The drug concentration dependent experiment commonly applies the drug at a log scale concentration dependent incubation in cells, instead of a linear scale concentration, such as Figure 2. Also, the range of the concentration needs to cover the IC50 of the drug, namely, including low concentration and high concentration to compare to the vehicle control. I would suggest the authors re-design the SNP concentration range to test in primary NSCs. For Fig 2B and 2D, all comparison should be pointed to the control, not the other drug concentration, thus, a Dunnett's test is preferred as the post-hoc analysis.

  Response: We greatly appreciate and agree with your suggestions. It’s better to apply the drug at a log scale concentration. However, the concentrations of SNP we used here are hard to distinct at a log scale, because the log value of 50, 100, 200, 300 and 400 are1.698, 2, 2.301, 2.477 and 2.602, respectively. IC50 of the drug is an important parameter pharmacology. We analyzed our original data, cell viability of 300 μM SNP-treated group is 48.154% of the control, which is very close to IC50. We will follow this suggestion to re-design the SNP concentration range to test in primary NSCs in our following experiments. The statistical analysis is corrected as “One-way ANOVA followed by Bonferroni multiple comparison tests was used to compare differences between means in more than two group” in this revision and marked with blue color.

Q4: Figure 3:

Same as Figure 2, log scale of SNP incubation in cells should be applied, instead of the linear scale in Fig 3 A, B.

Response: We greatly appreciate and agree with your suggestions. We will follow this suggestion to apply the drug at a log scale concentration of SNP in our following experiments.

For western blotting image presentation, although the authors are analyzing the ratio of p-p38 to p38, the internal control housekeeping gene expression should be shown in Fig 3 A, C.

Response: We are very sorry for mistake. The internal control housekeeping gene tubulin expression is shown in Fig 3 A, C.

  Importantly, the data in Fig 3 D and F graphs do not support the results described in Result section 3.3 Line 155-164. It shows that the pre-treatment with the p38 antagonist SB202190 has the same treatment effect as the SNP in increasing ratio of p-p38/p38 and decreasing OD value. Although the exact statistical significance is different, that does not change the interpretation of the results. Because p<0.05 was considered as significant, as described in Line 124. I would highly suggest the authors to test another p-38 antagonist, if the results are similar, then it may indicate that the SNP may not specifically work through the p-38 signaling in these NSCs.

Response: We appreciated very much for the professional suggestion. SB202190 is a pyridinyl imidazole derivative and is known to be a specific inhibitor of p38 MAPK [1, 2]. Peer researches showed SB202190 could inhibit p38 MAPK that activated by NO [2, 3]. In present study, we used SB202190 to inhibit p38 signaling pathway. It’s better to use another p-38 antagonist to make sure SNP specifically work through p38.

References:

  1. Yang, C., et al., A stress response p38 MAP kinase inhibitor SB202190 promoted TFEB/TFE3-dependent autophagy and lysosomal biogenesis independent of p38. Redox Biol, 2020. 32: p. 101445
  2. Kumar, A., et al., Inducible nitric oxide synthase is key to peroxynitrite-mediated, LPS-induced protein radical formation in murine microglial BV2 cells. Free Radic Biol Med, 2014. 73: p. 51-9.
  3. Turpeinen, T., et al., Dual specificity phosphatase 1 regulates human inducible nitric oxide synthase expression by p38 MAP kinase. Mediators Inflamm, 2011. 2011: p. 127587.

Q5: Figure 4:

Same as Figure 2, log scale of SNP incubation in cells should be applied, instead of the linear scale in Fig 4.

Response: We greatly appreciate and agree with your suggestions. We will follow this suggestion to apply the drug at a log scale concentration of SNP in our following experiments.

Q6: Author names: first name should be in front of the last name.

 Response: Following your kindly suggestion, the form of name is corrected in this revision.

Q7: When use the short name for nitric oxide, it should use capital letters “NO” instead of “no”. Please correct through the manuscript, such as Line 33: “Physiological amounts of no are required……” and others.

 Response: We are very sorry for mistake. The wrong writing of NO has been corrected.

Q8: The sentence is too long and difficult for readers to understand, please consider breaking into small sentences, such as Line 40-43

 Response: Following your kindly suggestion, the sentence is revised in this revision.

Q9: Please use the full name when first time introducing a new term, such as “nNOS” (Line 45), “iNOS” (Line 49)

  Response: Thanks you so much and we have corrected this typo, and carefully checked the manuscript to avoid similar issues.

Q10: References needed to be cited, such as Line 54, 55

  Response: Following your kindly suggestion, the referred references are cited in this revision.  

Q11: Grammar issues and typos needed to be corrected throughout the entire manuscript, such as Line 60-61, Line 131, Line 144

Response: Following your kindly suggestion, the mentioned sentences are corrected and marked with blue color in this revision.

Reviewer 3 Report

In this manuscript, the authors investigated the effect of sodium nitroprusside (SNP) on neuronal cell survival. Nitric oxide (NO) plays an important role in neuronal development; however, the mechanism of NO in neuronal development remains unclear. The authors investigated the effect of SNP on the differentiation of neural stem cells (NSCs). They showed that SNP increased apoptosis of NSCs, and that SB202190, a p38 MAPK inhibitor, prevented SNP-induced cell death. The authors should clarify the following points:

1. In Figure 4A, it appears that SNP inhibited the neurite outgrowth of NSCs. If SNPs do not affect differentiation, the authors should investigate their effect on neurite outgrowth.

2. Is SNP degraded in the media of NSCs, releasing NO?

3. There is an article with similar research content (PMID: 10899924). The authors should clarify the differences between this manuscript and the previous one.

4. Line 11: play → plays ?

Author Response

In this manuscript, the authors investigated the effect of sodium nitroprusside (SNP) on neuronal cell survival. Nitric oxide (NO) plays an important role in neuronal development; however, the mechanism of NO in neuronal development remains unclear. The authors investigated the effect of SNP on the differentiation of neural stem cells (NSCs). They showed that SNP increased apoptosis of NSCs, and that SB202190, a p38 MAPK inhibitor, prevented SNP-induced cell death. The authors should clarify the following points:

  1. In Figure 4A, it appears that SNP inhibited the neurite outgrowth of NSCs. If SNPs do not affect differentiation, the authors should investigate their effect on neurite outgrowth.

 Response: We appreciated very much for the professional suggestion. The effect of NO on neurite outgrowth is still controversy. Studies showed that NO could promote neurite outgrowth of neurons [1, 2], while some other considered NO could inhibit neurite outgrowth [3]. These might be caused by the different levels of NO. It is with great regret that we had not investigate their effect on neurite outgrowth in this study. We will follow this suggestion in our following experiments.

References:

  1. Cooke, R.M., et al., Nitric oxide synthesis and cGMP production is important for neurite growth and synapse remodeling after axotomy. J Neurosci, 2013. 33(13): p. 5626-37.
  2. Jiang, Y., et al., Near-infrared light-triggered NO release for spinal cord injury repair. Sci Adv, 2020. 6(39).
  3. Song, Y., et al., The Mechanosensitive Ion Channel Piezo Inhibits Axon Regeneration. Neuron, 2019. 102(2): p. 373-389 e6.
  4. Is SNP degraded in the media of NSCs, releasing NO?

 Response: Thank you for the question. SNP is a donor of the NO. It consists of an iron core surrounded by five cyanide ion molecules and one molecule of the nitrosonium ion (NO+) [1]. Its capability of producing NO seems to depend on its interaction with sulfhydryl-containing molecules [2]. These information is added introduction line 66-68 and marked with blue color in this revision.

References:

  1. Zoupa, E. and N. Pitsikas, The Nitric Oxide (NO) Donor Sodium Nitroprusside (SNP) and Its Potential for the Schizophrenia Therapy: Lights and Shadows. Molecules, 2021. 26(11).
  2. Grossi, L. and S. D'Angelo, Sodium nitroprusside: mechanism of NO release mediated by sulfhydryl-containing molecules. J Med Chem, 2005. 48(7): p. 2622-6.
  3. There is an article with similar research content (PMID: 10899924). The authors should clarify the differences between this manuscript and the previous one.

 Response: Many thanks to your kindly suggestion. We carefully compared the differences between the present study and the peer research mentioned here (PMID: 10899924). Peer research observed the death-enhancing effect of BDNF on NO used primary cultured cortical neurons. They found that BDNF markedly accelerated the NO induced death of neurons. The underlying mechanism might be related to p38 and ERK. The biggest difference between two studies is cell type selected. We used the differentiated NSCs which contained neural progenitor cells, neurons, astrocytes, and oligodendrocytes, which likes the early development stage of nerve system. Because the cells interact with each other in nervous system, we observed the effect of SNP on nervous system rather than a single cell type. The difference is discussed in this revision and marked with blue color.

  1. Line 11: play → plays ?

Response: We are sorry for the grammar error. The sentence is revised as “Nitric oxide (NO) is a crucial factor in regulating neuronal development” and marked with blue color

Round 2

Reviewer 1 Report

The authors replied correctly to the most important criticisms. However, the additional corrections and rephrasing are required:

Line 18 (Abstract): remove “and” at the beginning of the sentence: “And the phosphorylation of p38 was….”

Line 20: no changes for 50 μM concentration in comparison with the control (instead of “no changes in 50 μM in comparison with the control”)

Line 23: as it is rewritten it seems that the antagonist was pretreated (while the cells/NSCs were those that were pretreated); the antagonist was added; similarly, “after treatment” should be used (instead of “after being treated”)

Line 32: “NO pairs” is unclear

Line 48: resulting in the shift… (“in” was missing)

Line 78: the part “of cerebral cortex tissue brain” can be removes; it was already specified in the previous sentence that cortex was used

Line 97: following (instead of “followed”)

Line 100: were prepared (instead of “was”); pre-treated (instead of “pre-treating)

Line 104: if “and” is used, then the two sentences should be combined into one single sentence

Line 147: Harvested from tissue (“from” was missing)

Line 151: provide full name of GFAP

Line 161: phase contrast (instead of “contrast phase”)

Line 181: NSCs (instead of “NSC's”)

Line 244: another sentence that starts with “And” (?)

Line 246: unclear meaning of “that can destroy a great many biomolecules”

Line 247: add “processes” at the end of this sentence

Line 257: unclear meaning of “On account of the differentiated…”

Line 261: found that the (“that” was missing)

Line 274: correct the part “there did not have”

Author Response

The authors replied correctly to the most important criticisms. However, the additional corrections and rephrasing are required:

Response: Following your kindly suggestions, the typos and the grammatical errors have been carefully corrected (English editing by MDPI, Certificate-61186), and the missing information has been added in this revision.

Line 18 (Abstract): remove “and” at the beginning of the sentence: “And the phosphorylation of p38 was….”

Response: We appreciated your suggestion. We have been removed the word “and” in this sentence.

Line 20: no changes for 50 μM concentration in comparison with the control (instead of “no changes in 50 μM in comparison with the control”)

Response: We appreciated your suggestion. The sentence has been revised.

Line 23: as it is rewritten it seems that the antagonist was pretreated (while the cells/NSCs were those that were pretreated); the antagonist was added; similarly, “after treatment” should be used (instead of “after being treated”)

Response: Thank you for pointing these out, and we have corrected the referred sentences in this revision.

Line 32: “NO pairs” is unclear

Response: Thank you for pointing these out, and we have corrected in this revision.

Line 48: resulting in the shift… (“in” was missing)

Response: We are very sorry for mistake. The missing word “in” has been added in this revision.

Line 78: the part “of cerebral cortex tissue brain” can be removes; it was already specified in the previous sentence that cortex was used

Response: Following your kindly suggestion, the term “of cerebral cortex tissue brain” has been removed in this revision.

Line 97: following (instead of “followed”)

Response: Following your kindly suggestion, the word “followed” has been replaced by “following” in this revision.

Line 100: were prepared (instead of “was”); pre-treated (instead of “pre-treating)

Response: Following your kindly suggestion, the referred words have been corrected in this revision.

Line 104: if “and” is used, then the two sentences should be combined into one single sentence

Response: Thank you for pointing these out, and we have corrected the referred sentences in this revision.

Line 147: Harvested from tissue (“from” was missing)

Response: Thank you for pointing these out, the missing word has been added in this revision.

Line 151: provide full name of GFAP

Response: Thank you for pointing these out, the full name of GFAP has been provided in this revision.

Line 161: phase contrast (instead of “contrast phase”)

Response: Thank you for pointing these out, and the words have been corrected in this revision.

Line 181: NSCs (instead of “NSC's”)

Response: Thank you for pointing these out, and the word has been corrected in this revision.

Line 244: another sentence that starts with “And” (?)

Response: Thank you for pointing these out, and we have corrected the referred sentences in this revision.

Line 246: unclear meaning of “that can destroy a great many biomolecules”

Response: Following your kindly suggestion, the referred words have been corrected as “many types of “in this revision.

Line 247: add “processes” at the end of this sentence

Response: Following your kindly suggestion, the word “processes” have been added in this revision.

Line 257: unclear meaning of “On account of the differentiated…”

Line 261: found that the (“that” was missing)

Response: Thank you for pointing these out, the missing word has been added in this revision.

Line 274: correct the part “there did not have”

Response: Thank you for pointing these out, we have been corrected the sentence in this revision.

Reviewer 2 Report

I think the authors have addressed all my concerns to a certain degree. I appreciate their efforts. Thanks!

Author Response

Thank you for your kind comments.

Reviewer 3 Report

The authors have addressed the points which I noted.

Author Response

Thank you for your kind comments.